# A Change in Nosocomial Infections among Surgical ICU Patients in the COVID-19 Era and MALDI-TOF Mass Spectrometry—A Cross-Sectional Study

**DOI:** 10.3390/microorganisms12081510

**Published:** 2024-07-23

**Authors:** Leon Jedud, Ana Cicvarić, Maja Bogdan, Despoina Koulenti, Jordi Rello, Željko Debeljak, Kristina Kralik, Dario Mandic, Slavica Kvolik

**Affiliations:** 1Faculty of Medicine, Josip Juraj Strossmayer University of Osijek, 31000 Osijek, Croatia; leon.jedud@sb-insula.hr (L.J.); maja.bogdan@kbco.hr (M.B.); zeljko.debeljak@kbco.hr (Ž.D.); kkralik@mefos.hr (K.K.); dario.mandic@kbco.hr (D.M.); 2Insula County Special Hospital, 51280 Rab, Croatia; 3Department of Anesthesiology, Resuscitation and Critical Care, Osijek University Hospital, 31000 Osijek, Croatia; 4Department of Clinical Microbiology and Hospital Infections, Osijek University Hospital, 31000 Osijek, Croatia; 5Second Critical Care Department, Attikon University Hospital, Centre for Clinical Research, Faculty of Medicine, The University of Queensland, Brisbane 4029, Australia; despoina.koulenti@nhs.net; 6Centro de Investigacion Biomedica en Red de Enfermedades Respiratorias (CIBERES), Instituto de Salud Carlos III, 28029 Madrid, Spain; 7Osakidetza Basque Health Service, Donostia University Hospital, 20014 San Sebastián, Spain; 8Clinical Research/Epidemiology in Pneumonia and Sepsis (CRIPS), Vall d’Hebron Institute of Research (VHIR), 08035 Barcelona, Spain; 9Research FOREVA, CHU Nîmes, 34197 Nîmes, France; 10Department of Clinical Laboratory Diagnostics, Osijek University Hospital Center, 31000 Osijek, Croatia

**Keywords:** surgical intensive care, COVID-19 pandemic, MALDI-MS, microbiology, nosocomial infections, bacteria, enterobacteria, big data, critical care outcome

## Abstract

During the COVID-19 pandemic, changes occurred within the surgical patient population. An increase in the frequency of resistant Gram-negative bacteria has since been recorded worldwide. After the start of the COVID-19 pandemic, microbiological diagnostics in our institution was performed using MALDI-TOF mass spectrometry. With this study, we wanted to confirm whether it contributed to a greater number of pathogenic bacteria detected in surgical ICU patients. A total of 15,033 samples taken from 1781 surgical patients were compared during the period from 2016 to February 2020 and during the COVID-19 pandemic from March 2020 to February 2023. On patients’ admission, pathogenic bacteria were mostly isolated from the respiratory system (43.1% and 44.9%), followed by urine cultures (18.4 vs. 15.4%) before and during the pandemic. After the onset of the COVID-19 pandemic, there was a significant increase in the frequency of isolation of *Enterobacter* spp. (5.4 before vs. 9%, *p* = 0.014) and other enterobacteria (6.9 vs. 10.8%, *p* = 0.017) on patients’ admission to the ICU, respectively. Despite this change, mortality in the ICU during the post-COVID-19 period was reduced from 23 to 9.6% (*p* < 0.001). The frequency of bacterial isolation did not change with the application of MALDI-TOF technology. By identifying the microorganism while simultaneously recognizing some resistance genes, we were able to start targeted therapy earlier. With the application of other infection control methods, MALDI-TOF may have contributed to the reduction in mortality in surgical ICU patients during the COVID-19 pandemic.

## 1. Introduction

At the start of the COVID-19 pandemic, numerous changes took place in the organization of hospital admissions. Due to frequent infections and the need for testing, elective diagnostic and surgical procedures were postponed for many patients [1]. At the same time, more emergency patients were admitted to hospitals, especially to the ICU. Changes in the organization of staff working in the Intensive Care Unit (ICU) meant that many experienced nurses and doctors were redirected to the treatment of respiratory failure of COVID-19 patients [2]. They were replaced at the ICU by new staff members who often lacked experience in caring for critically ill patients on ventilators. Frequent staff changes and a higher proportion of emergency patients could result in a higher number of infections in ICUs [3].

Concurrently with these changes, awareness of the need for greater hand hygiene, wearing face masks and other protective equipment, separating patients, and other infection prevention and control measures has increased [4,5]. Due to the trend of increased prescribing of antibiotics during the COVID-19 pandemic and possible outbreaks of resistant strains, the need to reduce the use of antibiotics has been raised [6,7]. Among the consequences of infection with the SARS-COV-2 virus in the human gut microbiome, a decrease in the abundance of intestinal commensals, *Eubacterium ventriosum* and *Faecalibacterium prausnitzii*, was recorded [8]. At the same time, there was an increase in the frequency of isolation of opportunistic pathogens, such as the *Actinomyces viscosus* and *Clostridium hathewayi* strains [8]. A recent study by Brogna and colleagues confirmed that these alterations could be attributed to the bacteriophage potential of the SARS-CoV-2 virus [9]. All these changes may potentially contribute to a higher frequency of infections and thus a higher mortality in patients treated in the ICU.

Since May 2020, the Clinical Laboratory Diagnostic Department has begun using a new Matrix-Assisted Laser Desorption/Ionization-Time Of Flight (MALDI-TOF) mass spectrometry device that enables more specific and sensitive detection of microbiological agents and, in some cases, even antibiotics resistance factors [10,11]. The identification of pathogens is rapid, sensitive, and inexpensive [12].

Despite the changes in the human microbiome during the COVID-19 pandemic, more sensitive microorganism identification technologies, like MALDI-TOF, could give different results in the isolation of bacteria compared to standard methods [9,10,11]. It is also possible that improved identification of pathogens may help decrease ICU mortality. This study aimed to compare the results of microbiological swabs and patient survival in the periods before and during the COVID-19 pandemic in our surgical ICU.

## 2. Materials and Methods

### 2.1. Patients

Before the start of the investigation, permission was obtained from the Ethics Committee of Osijek University Hospital, Croatia, approval date 10 June 2022, reg. number 2158-61-46-22-124, Chairperson Prof. Ivan Pozgain. All patient information was anonymous, and only their registration number was used during the research. Since 2016, microbiological samples collected from the Department of Intensive Care Unit at the Clinical Hospital Osijek have been registered in a Microsoft Access (MsA) digital database. Microbiological findings were recorded from the beginning of data entry into the register, on 15 May 2016, until the data were analyzed on 2 February 2023. The MsA database contains information about the patient’s identity, date of birth, medical record number, the department or hospital where the patient was admitted from, admission and discharge dates from the ICU, length of mechanical ventilation, type of sample taken, isolated pathogen, and treatment outcome, i.e., information regarding ICU survival and discharge to surgical ward or home.

Each ICU stay was defined by an ICU registration number, which was the serial number of the ICU admission (i.e., 784/18). The same person could be recorded several times if they were discharged from the hospital and later readmitted, resulting in a new ICU registration number. Microbiological results read as “sterile” or “normal flora” were all considered negative. To display microbiological findings in a way that would provide a better insight into the causative agents isolated during a patient’s stay, all positive findings collected on the same day were displayed as “sets of samples”. For each patient stay, the first three sets of samples were displayed, in which at least one specimen was positive, provided that each set was sent on a different date. In this way, we can identify bacteria isolated on admission or during the ICU stay. After processing the data, we were able to see which bacteria prevailed upon admission and which prevailed after an extended ICU stay. Patients with all negative results were also included in the analyses.

### 2.2. Microbiology Analyses

In the pre-COVID-19 period, conventional microbial cultures were used for the identification of microorganisms. These methods were primarily based on biochemical tests (VITEK 2 microbial identification system, Boimerieux, France, or automated microbiology system Phoenix M50, Beckton Dickinson, Franklin Lakes, NJ, USA). After the start of the COVID-19 pandemic, all microbiological samples were analyzed using an ultrafleXtreme MALDI-TOF MS, Bruker, Bremen, Germany. Such technology was used to compare the peptide mass fingerprint with a database containing 334 Gram-negative bacteria, Gram-positive bacteria, and yeasts [12]. Along with the identification of bacteria, it enabled the simultaneous identification of some resistance genes ahead of susceptibility tests [13].

Microbiological specimens were classified into the following 8 categories: surveillance swabs, respiratory swabs, stool samples, blood cultures, cerebrospinal fluid samples, urine cultures, wound swabs, and swabs from body cavities. The stool sample was used only for detecting *Clostridium difficile* toxins; thus, if a stool sample had a “toxin negative, antigen positive” finding, it was considered negative for *C. difficile* in this study.

Microbiological isolates were grouped in the following 16 categories: *Acinetobacter* spp. (*A. baumanii*, *A. calcoaceticus*, *A. seifertii*), *Pseudomonas aeruginosa*, *Escherichia coli*, *Klebsiella* spp. *(K. pneumoniae*, *K. oxytoca*), *Enterobacter* spp. (*E. cloacae*, *E. kobei*, *E. aerogenes*, *E. species*), *other Enterobacteriae* (*Citrobacter diversus*, *Citrobacter freundii*, *Citrobacter koseri*, *Serratia marcescens*, *Morganella morganii*, *Proteus mirabilis*, *Proteus vulgaris*, *Raoultella ornithinolytica*), *Enterococcus* spp. (*E. faecalis*, *E. faecium*, *E. avium*, *Vancomycin-resistant Enterococcus*), *Staphylococcus aureus*, coagulase-negative *Staphylococci* (*CoN Staphylococcus*, *S. dysgalactiae*, *S. hominis*), *Streptococcus* spp. (*S. anginosus*, *S. pneumoniae*, *S. pyogenes*, *S. viridans*, *S. constellatus*, beta-hemolytic *Streptococcus group C*), anaerobes (*Clostridium bifermentans*, *Clostridium difficile*, *Peptostreptococcus* spp., *Prevotella buccae*, *Bacteroides ovatus*, *Bacteroides fragilis*), *Candida albicans*, *Candidae non-albicantes* (*C. famata*, *C. glabrata*, *C. guillermondii*, *C. kefyr*, *C. krusei*, *C. lusitaniae*, *C. metapsilosis*, *C. parapsilosis*, *C. tropicalis*), other fungi (*Aspergillus fumigatus*, *Saccharomyces cerevisiae*, *Trichosporon* spp.), skin and mucous membrane microbiota *(Corynebacterium* spp., *Neisseria flavescens*, *Pediococcus pentosaceus*, *Paenibacillus* spp., *Lactobacillus rhamnosus*, *Bacillus* spp., anthracoid *Bacilli*), and rare pathogens (*Aeromonas veronii*, *Stentotrophomonas maltophilia*, *Haemophilus influenzae*, *Leclercia adecarboxylata, Sphingomonas paucimobilis*).

### 2.3. Study Periods

Microorganisms that were confirmed and the types of positive samples from which they were isolated were compared in both pre-COVID-19 and COVID-19 periods. For this cross-sectional study, the pre-pandemic period from the beginning of the study in 2016 to February 2020 and the period of the COVID-19 pandemic from March 2020 to February 2023 were compared.

### 2.4. Statistical Analysis

Categorical data were represented by absolute and relative frequencies. The Chi-squared Test and Chi-squared test for trend (Cochran–Armitage test for trend) were used to analyze the differences in proportions between the studied samples. All P values were two-sided. The level of significance was set at an Alpha of 0.05. The statistical analysis was performed using MedCalc^®^ Statistical Software version 22.006 (MedCalc Software Ltd., Ostend, Belgium; https://www.medcalc.org; accessed on 10 July 2023)

## 3. Results

The data obtained from the MsA database consisted of 15,033 microbiological results that were organized into 1781 unique patients’ admissions in the ICU. In the period before the pandemic, 1233 surgical patients from whom samples were taken for microbiological analysis were admitted to the ICU. Of these samples, at least one pathogenic microorganism was isolated in 428 patients (34.7%). The frequency of isolation of bacteria in specific categories of microbiological samples is shown in Table 1.

Before the COVID-19 pandemic, the trend of a significant decrease in the frequency of isolation of pathogens from respiratory samples (χ^2^ test, *p* = 0.01) was observed in the second and third sets. During the same period, a trend of the significant increase in the frequency of isolation of pathogens from blood culture samples (χ^2^ test, *p* = 0.04) and wound swabs (χ^2^ test, *p* < 0.001) was observed in the second and then third set.

After the onset of the COVID-19 pandemic, a significant decrease in the frequency of isolation of pathogens from urine samples (χ^2^ test, *p* = 0.02) and respiratory samples (χ^2^ test, *p* = 0.02) was observed in the second and then third set. During the same period, a significant increase in the frequency of isolation of pathogens from surveillance swab samples (χ^2^ test, *p* = 0.03), wound swabs (χ^2^ test, *p* < 0.001), and cerebrospinal fluid samples (χ^2^ test, *p* = 0.03) was observed in the second and then third set (Table 1).

Groups of bacteria isolated in individual samples are shown in Table 2. By comparing bacteria in the first set of samples, a statistically significant difference was observed in the frequency of isolation of *Enterobacter* spp. (5.4 before vs. 9% after the onset of the COVID-19 pandemic, *p* = 0.014) and other enterobacteria (6.9 vs. 10.8%, *p* = 0.017). The frequency of other bacteria on admission did not differ between the two periods (Figure 1).

Before the COVID-19 pandemic, a trend was observed in which *Acinetobacter* spp. and Coagulase-negative Staphylococci were more frequently isolated in the second and then third set. At the same time, a significant decrease in the frequency of isolation of *E. coli*, *S. aureus*, *Enterobacter* spp., and *Klebsiella* spp. was observed in the second and third sets. During the COVID-19 pandemic, the isolation of *Acinetobacter* spp. in the second and third sets was more frequent than in the first set, but less frequent than in the period before the pandemic (*p* = 0.09). The isolation of *S. aureus* and *Enterobacter* spp. was less frequent in the second and third set of samples.

After the pandemic started, along with the changes mentioned above in the isolation of bacteria, the number of samples per patient taken during the treatment increased. At the same time, the number of patients admitted to the ICU decreased (Table 3). We compared these data during the observed years to see whether there were changes in treatment outcomes in the ICU, especially regarding patients with positive isolates.

## 4. Discussion

This study confirmed significant changes in the frequency of bacterial isolation in the surgical ICU during the COVID-19 pandemic compared to the period before it. The number of positive samples increased from 34.7 to 38%, which we believe was due to earlier detection and better diagnostics after the start of the COVID-19 pandemic. The most significant change is the increase in the frequency of the order *Enterobacterales* in the sample sets after the pandemic compared to the samples taken before the COVID-19 pandemic [7]. Despite that, a significant decrease in mortality over time compared to the period before the pandemic was observed after the onset of the COVID-19 pandemic. All these changes can be explained by a series of events that occurred during and after the COVID-19 pandemic.

With the onset of the pandemic, the admissions of surgical, especially cancer patients to the hospital decreased due to safety measures [14,15,16]. Also, the admission of elective thoracic, abdominal, and neurosurgical patients to surgical ICUs decreased worldwide due to due to less invasive surgical procedures and enhanced recovery after surgery (ERAS) protocols [17,18]. Consequently, most scheduled abdominal patients were referred to the surgical wards. In parallel with the decrease in the admission of elective surgical patients, the proportion of emergencies compared to elective surgical patients increased. Thus, in contrast to the pre-COVID-19 ratio of 60% elective and 40% emergency patients in our ICU, during the COVID-19 pandemic, emergency patients were predominant among those admitted to the ICU.

Neurosurgical, thoracic, abdominal, and other oncology patients admitted to the ICU during the pandemic were more often emergency patients. This can be partly explained by the lockdown measures to control the spread of the COVID-19 infection that were applied during the study period. Data on the delay in surgical treatment for patients with cancer were also observed in studies by other authors [16,19,20]. In the period after the COVID-19 pandemic, a higher comorbidity was also recorded in the general population, which particularly refers to a greater number of respiratory disorders. Consequently, Manuzzi and colleagues, in a study conducted in Lombardy after the COVID-19 pandemic, recorded a 32-fold increase in the use of chest CTs, 32 times more frequent spirometry, and 5.6 times more frequent ECG in ICU patients in the post-COVID-19 period [21].

Many factors can explain the data obtained in our study, the most important of which we consider early laboratory confirmation of inflammatory markers, microbiology diagnostics using MALDI-TOF technology, higher level of care after the start of the COVID-19 pandemic, and changes in the categories of patients admitted to the ICU.

The frequency of bacterial isolation in our study differs from other studies. In a systematic meta-analysis that included 38 studies with 2715 patients, Abu-Rub et al. confirmed that the frequency of bacterial infections was about 38% in patients with confirmed COVID-19 infections. The largest share was infections with staphylococcal strains at about 75%, which was attributed to the use of azithromycin and third-generation cephalosporin [22]. In our population, staphylococci were isolated less frequently, with a constant frequency of *S. aureus* and CoNS around 10% in all sets, both in pre- and post-COVID-19 era. On contrary, the frequency of G- bacteria from the genus *Enterobacterales* was significantly higher in the COVID-19 period, as observed in French Guiana by Lontsi Ngoula during the COVID-19 crisis [7]. They confirmed an increased risk of cefotaxime exposure with the emergence of extended-spectrum β-lactamase producing *Enterobacterales* [7].

With this study, we confirmed the initial frequency of pathogenic bacteria isolation, as well as their dynamics during the ICU treatment [23]. In our sample, the most common positive samples at admission were positive samples from the respiratory tract and positive urine cultures.

Positive samples from the respiratory tract were the most common even before the COVID-19 pandemic and, after the pandemic, their frequency at admission to the ICU increased from 43.1% to 44.9%. A higher frequency of respiratory infections during the COVID-19 pandemic was also observed by Castellarnau et al. [24]. The authors also noted a higher frequency of other complications in ICU patients, which they attributed to the impossibility of implementing the ERAS protocol [24].

The frequency of respiratory infections decreased significantly in samples taken later during ICU treatment. This was a consequence of recognition and early targeted antibiotic therapy. The average length of treatment in the ICU in our surgical patients with confirmed infection was 12 days, which was also observed in 2436 European ICU patients with nosocomial pneumonia [25]. This length of treatment has not changed over the years, and mortality has been significantly reduced, although emergency patients were predominant.

The reasons for this could be the application of new treatment methods such as hemofiltration, noninvasive ventilation, personalized clinical nutrition, and the introduction of physical therapists in the surgical ICU. Another parameter that may have improved patients’ outcomes was more extensive laboratory diagnostics, combined with microbiological diagnostics using MALDI-TOF, all aimed at the early detection of infections. By using MALDI-TOF, we confirmed the presence of bacteria that we had not found before [11,26]. These bacteria were earlier classified in a larger group and labeled as spp., such as *Sphingomonas paucimobilis*, *Delftia acidovorans*, or *Leclercia adecarboxylata*. The use of MALDI-TOF allowed us to assess both identification and resistance more precisely, which would be necessary for targeted and successful antibiotic therapy. Shortening the time of identification of microorganisms is one of the key factors in the rapid initiation of appropriate antimicrobial treatment. Like MALDI-TOF, multiplex PCR and other new methods can be used for the identification of positive samples, to speed up the application of targeted antimicrobial therapy after empirical treatment has been initiated [27,28].

The frequency of positive urine culture before the pandemic was 18.4%, and after the pandemic, its frequency decreased to 15.4% (ns, *p* = 0.201). These infections were commonly registered in the patients just upon ICU admission and were more efficiently eradicated during the subsequent ICU treatment than pulmonary infections [29]. This decrease could also be explained by the improved infection prevention measures, which, after the start of the COVID-19 pandemic, were undertaken in all segments of hospital treatment, starting with emergency admissions [30]. Considering that an increasing number of emergency patients were treated in ICUs, early prevention of infection might have contributed to the reduction in urinary tract infections, especially those related to catheter use [30].

Samples taken from patients admitted into the ICU showed a high frequency of *Acinetobacter* isolation before the pandemic and a higher frequency of isolation of *A. baumanii* in patients with prolonged ICU stays. After the COVID-19 pandemic, the frequency of isolation of resistant microorganisms from the order *Enterobacterales* increased [31,32]. This finding was also observed in other studies [33,34]. Their presence is currently becoming more frequent, and they are often recorded in patients upon admission to the hospital. Such categories are chronic patients, patients with neurological diseases or bedridden, and those who have previously used antibiotics. The implementation of antibiotic stewardship measures is important in the prevention of infections, but the emphasis on measures to reduce their transmission between patients must be equally important [32,33].

The results of the microbiological analyses that we have obtained point to possible improvements in clinical work. The increase in the frequency of isolation of pathogenic bacteria in surveillance swabs from the skin during the COVID-19 pandemic indicated the importance of hand washing, wearing protective equipment, isolating patients, and their hygiene. On the other hand, the largest number of wound infections was registered in abdominal patients. These infections were often caused by very resistant pathogens isolated from the abdominal cavity after emergency operations. An intraoperative sampling from the abdominal cavity can contribute to faster-targeted antibiotic therapy. Although this practice is uncommon and can prolong treatment in minor procedures, such as appendicitis, it can speed up the identification of the causative agent in high-risk patients with comorbidities [35,36]. Such conditions are incarcerated hernias, perforation of abdominal organs, surgical reinterventions, or conditions accompanied by ileus [36].

The perioperative empirical use of antibiotics in our institution is carried out according to the protocol and can be strongly related to the treatment outcome. Considering the growing number of resistant pathogens, it may not always be adequate. Krobot and colleagues have observed that in patients who underwent emergency abdominal surgery, the treatment was successful in 78.6% if their initial antibiotic treatment was appropriate. On the contrary, in those who were inappropriately treated, clinical success was registered in 53.4% of patients [37]. The length of hospital treatment also differed between these patients and was 13.9 days for those with the appropriate treatment vs. 19.8 days for those with inappropriate initial empiric antibiotics [37]. These data, as well as data on the increase in the frequency of isolation of resistant pathogens, emphasize the importance of prompt microbiological sampling, especially in high-risk surgical patients.

An increasing number of positive blood cultures in samples after admission to the ICU was observed before and during the COVID-19 pandemic in our surgical ICU. Bacteremia on admission to the ICU usually reflects sepsis due to acute surgical disease, e.g., sepsis with peritonitis due to bowel perforation. The expected causative agents are most often G- bacteria. The causative agents in hospital-acquired bloodstream infections (BSI) are often multi-resistant strains of G- bacteria such as *Acinetobacter*, *Klebsiella* spp., or Gram-positive bacteria. Hospital-acquired BSIs represent a target for future interventions. Possible interventions are limiting invasive vascular access to the shortest possible time, and taking hygiene measures, including hand hygiene, new type needleless intravenous connectors, disinfection hubs, etc. [38].

At the beginning of the pandemic, an increase in mortality was observed in our study. It was attributed to the impossibility of implementing the ERAS protocols, poor post-COVID-19 patients’ characteristics on admission, and the growing number of patients with advanced cancers and cancers with aggressive biology [24,39,40]. Due to the large number of patients admitted to the hospital as emergency surgical cases in the advanced stages of malignant disease, we have introduced palliative care measures. These were previously carried out on a case-by-case basis after consultation with doctors in charge [40]. At the level of the entire hospital, protocols have been established according to the instructions of the Ministry of Health. This is also one of the factors that contributed to the reduction in patient mortality in the ICU in previous years. These measures, including better microbiological diagnostics using MALDI-TOF and special attention aimed at preventing the spread of infections after the start of the COVID-19 pandemic, may play a role in better ICU patients’ outcomes that were observed in our study during the COVID-19 period. It is important to note that improved diagnostics alone cannot reduce the ICU mortality. In the study by Jeon et al. on 556 patients with bacteremia and/or fungemia, the authors examined pathogen identification, time to effective antibiotic therapy, time to microbiological clearance, length of ICU stay, 30-day mortality, and recurrence rate of the same BSI [41]. Without simultaneous antimicrobial stewardship, none of these parameters improved significantly [41].

Considering that the samples after the COVID-19 pandemic were analyzed using MALDI-TOF technology, MALDI-TOF may seem the main actor in this article. It is important to note that MALDI-TOF was not the only improvement in microbiological diagnostics after the start of the COVID-19 pandemic. BioFire respiratory panels, blood culture identification panels, and other tests for bacteria, viruses, and yeast have been introduced into diagnostics as well. These tests also enable the rapid identification of basic resistance genes, suggesting whether a favorable clinical response to applied empiric therapy is expected. Based on this, the preliminary written results were obtained before the susceptibility test.

Some limitations of this study must be mentioned too. Some data that may have a significant association with outcome were not analyzed, such as those on comorbidities, type of surgery, or antibiotic use. Prehospital use of antibiotics, which is also common and may be related to the outgrowth of resistant strains, was not recorded. Also, greater attention should be focused on confirming the existence of viral coinfection, which is especially common in mechanically ventilated ICU patients. In the study by Zelus et al., viruses were found in 42% of bronchoscopically obtained samples [28]. Patients with isolated viruses as a co-infection had a 10% higher mortality [28]. Unfortunately, in our study, such an analysis could not be performed because the presence of viruses, except for COVID-19, is not currently part of the routine infection screening in our institution.

## 5. Conclusions

In the period during the COVID-19 pandemic, bacteria from the *Enterobacterales* order were isolated more often compared to the pre-COVID-19 period. The frequency of pathogen isolation did not change, but bacteria were identified more precisely due to the introduction of MALDI-TOF technology; thus, there were more strains present that we had never previously identified. The use of MALDI-TOF technology helped in more precise identification and the earlier application of targeted antibiotic therapy. Better diagnostics with infection control procedures that were especially emphasized during the COVID-19 pandemic contributed to the reduction in mortality in our study.

## Figures and Tables

**Figure 1 microorganisms-12-01510-f001:**
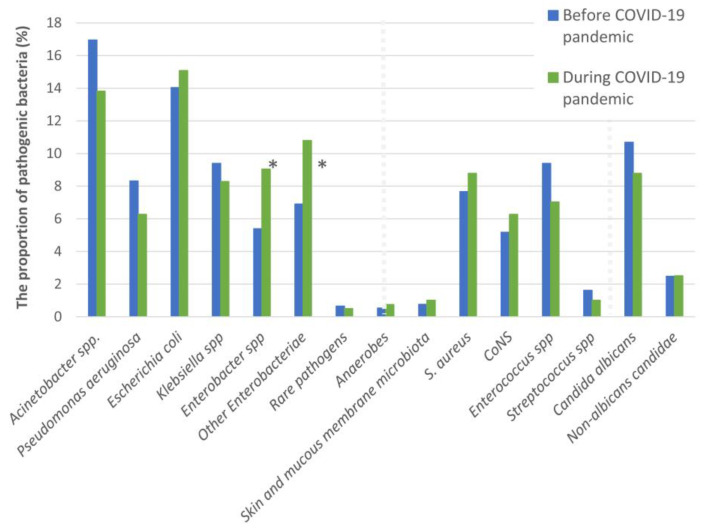
The frequency of isolation of pathogenic microorganisms in the time before and during the COVID-19 pandemic. G− bacteria are on the left side followed by G+ and yeasts, divided by a dotted line. Statistically significant differences in the first set of samples are shown with an asterisk (*).

**Table 1 microorganisms-12-01510-t001:** The frequency of isolation of individual pathogens to the total number of positive isolated pathogens in each set and the significance of the change within its period in the surgical ICU patients.

	Before COVID-19 Pandemic (1233 Patients Admitted)				During COVID-19 Pandemic (548 Patients Admitted)			
	1st Set	2nd Set	3rd Set		1st Set	2nd Set	3rd Set	
Admissions with isolated pathogens	428	374	227	**<0.001**	222	156	81	**0.006**
Swabs with isolated pathogens	902 (2.1/pt)	448 (1.2/pt)	284 (1.3/pt)		376 (1.7/pt)	177 (1.1/pt)	95 (1.2/pt)	
Swab type ***	1. set	2. set	3. set	*p* †	1. set	2. set	3. set	*p* †
Skin samples (surveillance)	64 (7.1)	33 (7.4)	24 (8.5)	0.48	19 (5.1)	11 (6.2)	11 (11.6)	**0.03**
Respiratory	389 (43.1)	169 (37.7)	101 (35.6)	**0.01**	169 (44.9)	62 (35)	33 (34.7)	**0.02**
Stool	4 (0.4)	3 (0.7)	3 (1.1)	0.25	0	3 (1.7)	0	0.31
Blood culture	70 (7.8)	44 (9.8)	33 (11.6)	**0.04**	35 (9.3)	19 (10.7)	11 (11.6)	0.46
Cerebrospinal fluid	11 (1.2)	6 (1.3)	4 (1.4)	0.79	5 (1.3)	8 (4.5)	4 (4.2)	**0.03**
Urine culture	166 (18.4)	77 (17.2)	40 (14.1)	0.10	58 (15.4)	24 (13.6)	5 (5.3)	**0.02**
Wounds	62 (6.9)	67 (15)	43 (15.1)	**<0.001**	29 (7.7)	30 (16.9)	17 (17.9)	**<0.001**
Body cavities	136 (15.1)	49 (10.9)	36 (12.7)	0.12	61 (16.2)	20 (11.3)	14 (14.7)	0.38

* The percentage of positive swabs calculated to the total number of samples with isolated pathogens in each set of samples. † Chi-squared test for trend (Cochran–Armitage test for trend). Statistically significant differences are bolded.

**Table 2 microorganisms-12-01510-t002:** The frequency of isolation of individual pathogenic bacteria in each set and the significance of the change between individual sets of samples before and after the start of the COVID-19 pandemic.

	Before Pandemic (1233 Patients Admitted)				During Pandemic (548 Patients Admitted)			
	1st Set	2nd Set	3rd Set	*p* †	1st Set	2nd Set	3rd Set	*p* †
Admissions with isolated pathogens	428	374	227	**<0.001**	222	156	81	**0.001**
Isolated pathogens ***	926 (2.2/set)	470 (1.3/set)	298 (1.3/set)		398 (1.8/set)	181 (1.2/set)	92 (1.3/set)	
*Acinetobacter* spp.	157 (17)	138 (29.4)	106 (35.6)	**<0.001**	55 (13.8)	35 (19.3)	24 (26.1)	**0.003**
*Pseudomonas aeruginosa*	77 (8.3)	51 (10.9)	29 (9.7)	0.26	25 (6.3)	12 (6.6)	12 (13)	0.06
*Escherichia coli*	130 (14)	38 (8.1)	13 (4.4)	**<0.001**	60 (15.1)	21 (11.6)	12 (13)	0.39
*Klebsiella* spp.	87 (9.4)	34 (7.2)	14 (4.7)	**0.007**	33 (8.3)	10 (5.5)	9 (9.8)	0.95
*Enterobacter* spp.	50 (5.4)	17 (3.6)	7 (2.3)	**0.02**	36 (9)	10 (5.5)	2 (2.2)	**0.01**
Other *enterobacteriae*	64 (6.9)	34 (7.2)	23 (7.7)	0.63	43 (10.8)	16 (8.8)	9 (9.8)	0.60
Rare pathogens	6 (0.6)	1 (0.2)	2 (0.7)	0.77	2 (0.5)	3 (1.7)	1 (1.1)	0.32
Anaerobes	5 (0.5)	5 (1.1)	4 (1.3)	0.14	3 (0.8)	3 (1.7)	0	**0.004**
Skin and mucous membrane microbiota	7 (0.8)	5 (1.1)	0	0.33	4 (1)	0	0	0.13
*Staphylococcus aureus*	71 (7.7)	25 (5.3)	11 (3.7)	**0.008**	35 (8.8)	7 (3.9)	2 (2.2)	**0.005**
CoNS	48 (5.2)	25 (5.3)	28 (9.4)	**0.02**	25 (6.3)	26 (14.4)	7 (7.6)	0.11
*Enterococcus* spp.	87 (9.4)	32 (6.8)	23 (7.7)	0.19	28 (7)	13 (7.2)	5 (5.4)	0.67
*Streptococcus* spp.	15 (1.6)	5 (1.1)	1 (0.3)	0.07	4 (1)	0	0	0.13
*Candida albicans*	99 (10.7)	40 (8.5)	29 (9.7)	0.41	35 (8.8)	20 (11)	7 (7.6)	0.96
*Candidae non-albicantes*	23 (2.5)	19 (4)	7 (2.3)	0.68	10 (2.5)	3 (1.7)	2 (2.2)	0.68
Other fungi	0	1 (0.2)	1 (0.3)	0.11	0	2 (1.1)	0	0.37

* The ratio of bacteria was calculated to the total number of isolated bacterial strains in each set. † Chi-squared test for trend (Cochran–Armitage test for trend). CoNS—Coagulase-negative *Staphylococci.* Statistically significant differences are bolded.

**Table 3 microorganisms-12-01510-t003:** Comparison of the number of patients, mortality, length of ICU stays, and length of mechanical ventilation in all patients treated in the ICU compared to patients with at least one positive microbiological finding.

Year	Total ICU Stays	Positive ICU Stays	ICU Mortality	ICU Mortality of Positive Stays	Days of ICU Stay	Length of MV (Days)	Days of Positive Stays	Length of MV in Positive Stays (Days)
2017	819	196 (24%)	196 (24%)	98 (50%)	4.04	2.25	11.8	7.63
2018	793	166 (21%)	137 (17%)	71 (43%)	4	2.19	12.63	8.51
2019	759	224 (30%)	111 (15%)	58 (26%)	4.48	2.63	11.4	7.76
2020	575	141 (25%)	103 (18%)	50 (35%)	4.58	2.86	12.29	9.05
2021	572	99 (17%)	55 (10%)	17 (17%)	3.68	2.16	12.13	8.77
2022	625	128 (20%)	60 (9.6%)	35 (27.3%)	4	2.18	12.8	8.6
*p* *		0.001	<0.001	<0.001	0.929	0.967	0.312.	0.402

* χ^2^ test, ICU intensive care unit, MV mechanical ventilation.

## Data Availability

The data presented in this study are available on request from the corresponding author.

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
