# Peer review of "A Change in Nosocomial Infections among Surgical ICU Patients in the COVID-19 Era and MALDI-TOF Mass Spectrometry—A Cross-Sectional Study"

_microorganisms, 2024, doi:10.3390/microorganisms12081510_

Round 1

Reviewer 1 Report

Comments and Suggestions for Authors

Authors Kvolik et al. describe the use of mass spectrometry to detect bacterial pathogens during COVID-19 infection. It is a well-articulated work. I request the authors to better expand the introduction by citing work that has used similar techniques and similarly to do in the discussion. Also, the authors should argue a fact that it is now known that SARS-CoV-2 has bacteriophage behavior. I ask that the abbreviations be revised and that a more ample description of their technique be introduced in the materials and methods

Author Response

Dear Reviewer 1
Our comments are attached

Prof. Kvolik 

Reviewer 2 Report

Comments and Suggestions for Authors

Manuscript A change of nosocomial infections among surgical ICU patients in the COVID-19 era & MALDI TOF mass spectrometry, a cross-sectional study by Jedud et al deals with infections in intensive care unite and its correlations to COVID-19 pandemics. The paper is written nicely, all aspects that should be mentioned are covered, and discussed in details which allow the complete overview of the topic. 

I have no specific comments/questions/suggestions for Authors, except minor ones>

1. Please add the Consent of Ethical Committee for the study.

2. Please check- if stating, eg. this was seen in studies ....and then there is only one reference, please put singular et vice versa.

3. Some minor spelling mistakes are seen in Methods, please check them.

Best of wishes in publishing your paper!

Author Response

Dear Reviewer 2
Our comments are attached

Thank you for the useful suggestions 

Prof. Kvolik 

Reviewer 3 Report

Comments and Suggestions for Authors

Comments and suggestions

Summary section:

1. What is the relevance of mentioning malignant neoplasms?

2. Are the study periods before, during, or after the pandemic?

Introduction section:

3. Not only focus on hand washing, mention the other factors that have allowed infection control.

4. Are the objectives focused on comparing the results of swabs and survival? But is this comparable? or are you looking to establish a relationship?

Materials and methods section:

5. Indicate what you mean by saying that the database has patient identity information...when this should be anonymous.

6. Indicate the type of study

7. It is understood that it is an ambispective study. Retrospective for the data before the pandemic and prospective for the data during the pandemic but this is not specified nor is it clear in the methodological section.

8. Because a control group was not considered

Results section:

9. It is not clear how the groups have been divided

10. The conclusion is general and redundant. Restructure

Comments on the Quality of English Language

 Minor editing of English language required

Author Response

Dear Reviewer 3
Our comments are attached

Thank you for the useful suggestions 

Prof. Kvolik 

Round 2

Reviewer 3 Report

Comments and Suggestions for Authors  1. It has not been possible to justify or place limitations on the absence of a control group in the study. 2. The methodological section is not clear. In points 6 and 7 of the response letter they indicate that the design is transversal as indicated by their statesman. However, in point 8 they indicate that the study or part of it was retrospective.

For this reason, in my opinion, it should not be considered for publication without a general restructuring of the manuscript.

}Resultados de traducción

Resultado de traducció

Comments on the Quality of English Language

Minor editing of English language required
